# Association Analysis of *PRKAA2* and *MSMB* Polymorphisms and Growth Traits of Xiangsu Hybrid Pigs

**DOI:** 10.3390/genes14010113

**Published:** 2022-12-30

**Authors:** Jiali Xu, Yong Ruan, Jinkui Sun, Pengfei Shi, Jiajin Huang, Lingang Dai, Meimei Xiao, Houqiang Xu

**Affiliations:** 1Key Laboratory of Animal Genetics, Breeding and Reproduction in the Plateau Mountainous Region, Ministry of Education, Guizhou University, Guiyang 550025, China; 2Guizhou Provincial Key Laboratory of Animal Genetics, Breeding and Reproduction, Guizhou University, Guiyang 550025, China; 3College of Animal Science, Guizhou University, Guiyang 550025, China

**Keywords:** *PRKAA2*, *MSMB*, SNPs, real-time PCR, Xiangsu hybrid pigs, growth traits

## Abstract

In this study, Xiangsu hybrid pig growth traits were evaluated via *PRKAA2* and *MSMB* as candidate genes. Sanger sequencing revealed three mutation sites in *PRKAA2*, namely, g.42101G>T, g.60146A>T, and g.61455G>A, and all these sites were intronic mutations. Moreover, six mutation sites were identified in *MSMB*: intronic g.4374G>T, exonic g.4564T>C, exonic g.6378G>A, exonic g.6386C>T, intronic g.8643G>A, and intronic g.8857A>G. Association analysis revealed that g.42101G>T, g.60146A>T, g.61455G>A, g.4374G>T, g.4564T>C, g.6378G>A, g.6386C>T, g.8643G>A, and g.8857A>G showed different relationship patterns among body weight, body length, body height, chest circumference, abdominal circumference, tube circumference, and chest depth. Real-time polymerase chain reaction results revealed that the expression of *PRKAA2* was highest in the longissimus dorsi muscle, followed by that in the heart, kidney, liver, lung, and spleen. The expression of *MSMB* was highest in the spleen, followed by that in the liver, kidney, lung, heart, and longissimus dorsi muscle. These results suggest that *PRKAA2* and *MSMB* can be used in marker-assisted selection to improve growth related traits in Xiangsu hybrid pigs, providing new candidate genes for Pig molecular breeding.

## 1. Introduction

The Xiangsu hybrid pig is a cross between the Guizhou Congjiang Xiang pig and Jiangsu Sutai pig of China. The Xiang Pig, a small pig breed unique to the Guizhou Province in China, is characterized by early sexual maturity, better disease resistance, and strong adaptability than other local breeds [1,2,3]. It is highly homozygous in its genome composition and highly similar to the human genome. It is considered an ideal animal model for toxicology and pharmacology research [4,5,6,7,8]. The Congjiang Xiang pigs have many advantages, but they grow slowly. Therefore, in the early stages of research, Congjiang Xiang pigs and Sutai pigs were crossed, and their offspring were backcrossed. The blood relationship of Congjiang Xiang pigs accounted for 87.5% in the offspring. In this sudy, *PRKAA2* and *MSMB* genes were used as candidate genes for growth traits, and the relationships among different genotypes in these two candidate genes and growth traits at six months of age were analyzed by Sanger sequencing to screen for superior genes or superior growth trait genotypes.

*PRKAA2* (adenosine monophosphate-activated protein kinase (AMPK)) is a member of the AMPK family. They are heterotrimeric proteins that mainly detect the state of mammalian cells, regulate the new biosynthesis of fatty acids and cholesterol, and play a key role in cell energy metabolism [9,10,11]. A previous study found that *PRKAA2* is primarily expressed and distributed in the heart, skeletal muscle, liver, and neuronal tissue cells of animals [12,13,14]. In clinical research, *PRKAA2* has been shown to be associated with susceptibility to type 2 diabetes [15,16], In addition, the high expression of *PRKAA2* may predict the poor prognosis of head and neck squamous cell carcinoma [17], colorectal cancer [18], and endometrial cancer [19]. In animal research, the bovine *PRKAA2* was located on chromosome 3 using somatic cell hybrid cell panel [20]. The study on *PRKAA2* in Pakistani Nili-Ravi and Kundi buffaloes revealed 17 single nucleotide polymorphism (SNP) loci. These SNPs may be related to energy metabolism and production traits [21]. *PRKAA2* was also studied in Qinchuan cattle, Nanyang cattle, Jiaxian cattle, and yak cattle. The phylogenetic analysis revealed that Qinchuan cattle and Jiaxian cattle are closely related, followed by Nanyang cattle, and yak cattle has the largest relationship difference, which provided evolutionary information for the genetic background of different cattle breeds in China [22]. However, few studies have evaluated *PRKAA2* in pigs.

*MSMB* (microseminoproteinp-β) is a disulfide rich low molecular weight protein synthesized by prostaglandin epithelial cells. This protein inhibits the growth of prostate tissue via the autocrine and paracrine modes. *MSMB* has been reported to have a relatively low sequence identity in vertebrates but exhibit strong conservation [23,24,25]. In clinical studies, *MSMB* was mainly associated with prostate cancer treatment and other aspects [26,27]. In animals, it has been mainly associated with reproduction [28,29,30] and not linked to animal growth, particularly in pigs. *MSMB* was selected as a candidate gene for evaluating growth traits in this study. Thus, *PRKAA2* and *MSMB* were selected as candidate genes for evaluating growth traits to identify dominant genes or dominant genotypes in Xiangsu hybrid pigs.

## 2. Materials and Methods

### 2.1. Ethical Approval

All experiments in this study were performed in accordance with the guidelines of the Animal Welfare Committee of Guizhou University (EAE-GZU-2021-P003, 9 May 2021).

### 2.2. Bioinformatics Analyses

The amino acid sequences of *PRKAA2* and *MSMB* were obtained from NCBI (https://www.ncbi.nlm.nih.gov/protein accessed on 6 November 2021) for *Sus scrofa* (NP_999431.1; NP_999017.1), *Bos taurus* (NP_001192534.1; XP_024842536.1), *Bubalus bubalis* (XP_045022084.1; XP_006065751.3), *Capra hircus* (XP_017900141.1; XP_013831148.2), *Ovis arie* (NP_001106287.1; XP_042096743.1), *Equus caballus* (NP_001075410.1; XP_014589601.1), *Equus asinus* (XP_044626814.1; XP_044615142.1), *Gallus gallus* (NP_001034694.1; XP_004942175.3), *Anser cygnoides* (XP_047918049.1; XP_013033273.1), and *Oryctolagus cuniculus* (XP_008263418.1; XP_008268062.2). MEGA 7 (https://www.megasoftware.net/ accessed on 12 October 2021) was used for sequence alignment and phylogenetic tree construction. The structural features and functions of *PRKAA2* and *MSMB* proteins in these 10 species were revealed using MEME (https://meme.nbcr.net/ accessed on 12 October 2021) suite software.

### 2.3. Animal Weight and Body Size Data and Tissue Collection

A total of 164 Xiangsu hybrid pigs for this study were provided by the Xiang pig breeding farm of Guizhou University. Their blood samples were collected, and their growth traits (body weight, body length, body height, chest circumference, abdominal circumference, tube circumference, chest depth, chest width, and leg and hip circumference) at the age of 6 months were measured for correlation analysis. Three 6-month-old Xiangsu hybrid pigs were sacrificed, and six tissue samples of the heart, liver, spleen, lung, kidney, and longissimus dorsi muscle were collected for subsequent experiments.

### 2.4. DNA and RNA Extraction and cDNA Synthesis

DNA was extracted from the 164 pig blood samples according to the instructions of Omega blood genomic DNA extraction kit (D3392-01, GA, USA). Total RNA was extracted from six tissue samples using Tiangen RNA extraction kit (DP424, Beijing, China). cDNA was then synthesized using Genstar reverse transcription kit (A212-02, Beijing, China). The concentration and purity of DNA, RNA, and cDNA were measured via Thermo Nanodrop 2000 ultramicro spectrophotometer (NanoDrop 2000, CA, USA) and then stored at −20 °C.

### 2.5. Primer Design

Using the NCBI data of porcine *PRKAA2* (NC_010448.4) and *MSMB* (NC_010456.5) (https://www.ncbi.nlm.nih.gov/ accessed on 11 September 2020 ) as a reference, Premier 5.0 software was used to design relevant primers (Table 1 and Table 2). The primers were manufactured by Tsingke Biotechnology Co., Ltd. (Beijing, China).

### 2.6. Polymerase Chain Reaction (PCR)

The PCR reaction mixture (30 μL) contained the following components: 15 μL 2 × Taq PCR Starmix (GeneStar, Beijing, China), 10.5 μL ddH_2_O, 1.5 μL DNA template (40 ng/μL), and 1.5 μL each of the forward and reverse primers. The PCR amplification procedure was as follows: 94 °C for 5 min; 35 cycles of denaturation at 94 °C for 10 s, gradient annealing of 55–65 °C (55 °C, 57 °C, 59 °C, 61 °C and 63 °C) for 20 s, and extension temperature for 30 s; and finally holding at 72 °C for 5 min. The PCR products were subjected to agarose gel electrophoresis for 35 min. A gel imaging system (Universal Hood II, Bio–Rad, USA) was used to identify the bright bands and determine whether the length of the target fragment was consistent.

### 2.7. Real-Time PCR

The Real-time PCR reaction mixture (10 μL) included 5 μL SsoFastTM EvaGreen^®^ Supermix (Bio-Rad, CA, USA), 0.5 μL cDNA template, 0.5 μL each of the primers, and 3.5 μL ddH_2_O. The quantitative real-time PCR amplification procedure was as follows: predenaturation at 95 °C for 30 s, 39 cycles of denaturation at 95 °C for 5 s, gradient annealing from 55 °C to 65 °C for 5 s, and extension for 30 s, and final extended at 95 °C for 5 s.

### 2.8. Data Analysis

Microsoft Excel was used to calculate the gene frequency, genotype frequency, genetic homogeneity, genetic heterogeneity, effective allele number, and polymorphic information content (PIC). The χ^2^ test and *p* values were used to determine whether the allele frequency of the mutant group complied with the Hardy–Weinberg equilibrium. SPSS 25 software was used to perform correlation analyses for the body weight, body length, body height, chest circumference, abdominal circumference, tube circumference, chest depth, chest width, and leg hip circumference of the different genotypes, and the results are expressed as the mean ± standard deviation.

The 2^−∆∆Ct^ method was used to calculate the results of real-time PCR. The test data were analyzed using one-way analysis of variance in SPSS 25 and visualized using GraphPad Prism 8 (version 8, Harvey Motulsky, CA, USA).

## 3. Results

### 3.1. Bioinformatic Analysis of PRKAA2 and MSMB

The multiple sequence alignments of *PRKAA2* and *MSMB* proteins were performed with ten mammalian or domestic animals viz: pig (*S. scrofa*), cattle (*B. taurus*), buffalo (*B. bubalis*), goat (*C. hircus*), sheep (*O. aries*), horse (*E. caballus*), donkey (*E. asinus*), chicken (*G. gallus*), goose (*A. cygnoides*), and rabbit (*O. cuniculus*). As per the *PRKAA2* evolutionary tree, sheep and horses are closely related and pigs and cattle are closely related (Figure 1a). A total of 17 significant motifs were found in the 10 species (Figure 1b,c). According to the *MSMB* evolutionary tree, buffalo and goat are closely related and horse and rabbit are closely related (Figure 1d). Four significant motifs were found in the 10 species (Figure 1e,f).

### 3.2. Detection of SNP Mutation Sites in PRKAA2 and MSMB

#### 3.2.1. PRKAA2 SNP Mutation Site Detection

The porcine *PRKAA2* is located on chromosome 6 and contains nine exon regions. Sanger sequencing showed three mutation sites in *PRKAA2*, including intronic g.42101G>T, intronic g.60146A>T, and intronic g.61455G>A (Figure 2).

#### 3.2.2. MSMB SNP Mutation Site Detection

The porcine *MSMB* is located on chromosome 14 and include four exon regions. Sanger sequencing revealed nine mutation sites in *MSMB*, including intronic g.4374G>T, exonic g.4564T>C, exonic g.6378G>A, exonic g.6386C>T, intronic g.8643G>A, and intronic g.8857A>G (Figure 3).

### 3.3. Genetic Diversity Analysis of PRKAA2 and MSMB Mutation Groups

#### 3.3.1. Genetic Diversity Analysis of PRKAA2 Mutant Groups

Table 3 shows the genotype frequency and genetic parameters of the three mutant positions of the porcine *PRKAA2.* Their PIC was 0.374, 0.365, and 0.356 (all 0.25 < PIC < 0.5), indicating moderate polymorphism. The χ^2^ results showed that the three sites were in Hardy–Weinberg equilibrium.

#### 3.3.2. Genetic Diversity Analysis of MSMB Mutant Groups

Table 4 shows the genotype frequency and genetic parameters of the nine mutant positions of the porcine *MSMB*. According to the sequencing results, in the g.6378G>A site and g.6386C>T site, the mutant individuals and number of mutations were the same. PIC for all sites were in the range of 0.25 < PIC < 0.5, suggesting moderate polymorphism. The χ^2^ results showed that the six sites were in Hardy–Weinberg equilibrium.

### 3.4. Correlation Analysis of Genotypes and Growth Traits

To screen the relationship between the SNPs and growth traits of Xiangsu hybrid pigs, SPSS software was used to analyze the association between *PRKAA2* and *MSMB* mutation sites and nine traits (Table 5 and Table 6). In *PRKAA2*, the g.42101G>T locus was associated with body weight, body length, chest circumference, abdominal circumference, and tube circumference. the g.60146A>T locus was associated with chest depth; and the g.61455G>A locus was associated with body weight and chest depth. In *MSMB*, the g.4374G>T locus was associated with body weight, body height, and chest depth; the g.4564T>C locus was associated with body height and chest depth; the g.6378G>A and g.6386C>T loci were associated with chest depth; and the g.8643G>A and g.8857A>G loci were associated with body height and abdominal circumference.

### 3.5. Expression of PRKAA2 and MSMB in Different Tissues

The real-time PCR results are presented using the heart as the reference tissue (Figure 4). In the expression of *PRKAA2* in various tissues, there was a significant difference between the heart and liver, spleen, lung, and kidney (*p* < 0.01) but no significant difference between the heart and longissimus dorsi muscle (*p* > 0.05). The specific expression level from high to low was as follows: longissimus dorsi muscle > heart > kidney > liver > lung > spleen. In the expression of *MSMB* in various tissues, there was a significant difference between the heart and liver, spleen, lung, kidney, and longissimus dorsi muscle (*p* < 0.01). The specific expression level from high to low was as follows: spleen>liver>kidney>lung>heart>longissimus dorsi muscle.

## 4. Discussion

In the present study, intronic mutations were detected in *PRKAA2* and *MSMB*. Although an intron is a protein noncoding sequence, studies have found that intron variation can affect the expression and transcription of eukaryotic genes, RNA stability, processing of transcripts, efficiency of mRNA transport and translation from the nucleus to cytoplasm, and detection and elimination of mRNA errors in the transcription process [31,32,33,34]. Therefore, intronic mutations may also affect animal growth traits and be used as gene markers.

A total of three SNPs were identified in the *PRKAA2* of the Xiangsu hybrid pig population. Correlation analysis showed that the g.42101G>T locus was correlated with body weight, body length, chest circumference, abdominal circumference, and tube circumference (*p* < 0.05, *p* < 0.01); the g.60146A>T locus was correlated with chest depth (*p* < 0.01); and the g.61455G>A locus was associated with body weight and chest depth (*p* < 0.01). Therefore, g.42101G>T and g.61455G>A may be used as candidate sites for body weight and g.60146A>T and g.61455G>A may be used as candidate sites for chest depth. However, correlation between *PRKAA2* and Hu sheep and Dorper sheep growth revealed a mutation site at 32832382 (namely 32832382G>A locus), and association analysis with growth traits showed that individuals with the AA genotype had significantly higher body weight, body length, chest circumference, and gun circumference than those with the GG and GA genotypes (*p* < 0.05) [35]. In another study on the correlation between *PRKAA2* and the meat quality traits of commercial pigs, it was found that the statistical score of the physical location of the *PRKAA2* locus was not significant, thereby not necessitating any follow-up analysis [36]. Real-time PCR results showed that the expression of *PRKAA2* was highest in the longissimus dorsi muscle, followed by the heart, kidney, liver, lung, and spleen. The results of *PRKAA2* expression in sheep tissues showed that *PRKAA2* was expressed in the heart, liver, spleen, lung, kidney, rumen, duodenum, muscle, lymph, and tail fat, with the highest expression in the spleen, followed by the kidney, duodenum, and muscle [35]. These results suggest that the expression pattern of *PRKAA2* is different in different species and is species-specific.

Regarding *MSMB*, six SNPs were detected in the Xiangsu hybrid pig population. The analysis results showed that the g.4374G>T locus was correlated with body weight, body height, and chest depth (*p* < 0.05, *p* < 0.01); the g.4564T>C locus was correlated with body height and chest depth (*p* < 0.05, *p* < 0.01), the g.6378G>A and g.6386C>T loci were correlated with chest depth (*p* < 0.01), and the g.8643G>A and g.8857A>G loci were correlated with body height and abdominal circumference (*p* < 0.01). Therefore, g.4374G>T, g.4564T>C, g.8643G>A, and g.8857A>G may be used as candidate sites for body height, whereas g.4374G>T, g.4564T>C, g.6378G>A, and g.6386C>T may be used as candidate sites for body height. In addition, g.8643G>A and g.8857A>G may be used as candidate sites for abdominal circumference. The T allele of rs10993994 is a potential pathogenic variant of 10q11 that increases the risk of prostate cancer [37,38] and is an important marker for prostate cancer therapeutic targets [39]. Real-time PCR results revealed that the expression of *MSMB* in the spleen was the highest, followed by the liver, kidney, lung, heart, and longissimus dorsi muscle. The *MSMB* gene contains one pseudogene and four functional genes in Callithrix jacchus. Real-time RT-PCR analysis of RNA samples from the prostate, seminal vesicles, and testes of marmosets showed transcripts for *MSMB1*, *MSMB2*, *MSMB3*, and *MSMB4,* four functional genes, present in the testes and parotoid gonads, and levels in the prostate were approximately 100-fold higher than those in the seminal vesicles and 10,000-fold higher than those in the testes [25]. At present, there is no report on the expression of *MSMB* in animal tissues.

## 5. Conclusions

In the present study, Sanger sequencing was used to identify three mutation sites in *PRKAA2* and six mutation sites in *MSMB*. The late correlation analysis with growth traits revealed that *PRKAA2* was closely related to body weight, body length, chest circumference, abdominal circumference, tube circumference, and chest depth and *MSMB* was closely related to body weight, body height, abdominal circumference, and chest depth. These results suggest that *PRKAA2* and *MSMB* SNPs may be used as genetic markers for breeding Xiangsu hybrid pigs with desirable growth-related traits, which will provide new candidate genes for the molecular breeding of pigs in the future. The real-time PCR results of *PRKAA2* expression from high to low were: longissimus dorsi>heart>kidney>liver>lung>spleen, whereas those of *MSMB* expression from high to low were: spleen>liver>kidney>lung>heart>longissimus dorsi muscle.

## Figures and Tables

**Figure 1 genes-14-00113-f001:**
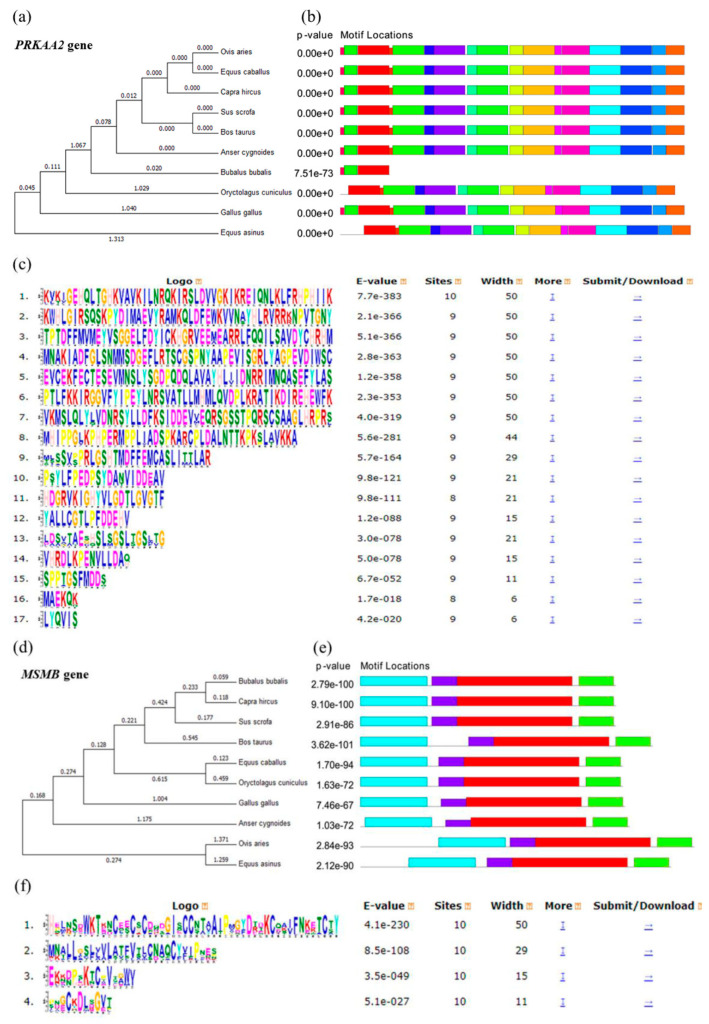
Bioinformatics analysis results of *PRKAA2* and *MSMB*. (**a**–**c**) are *PRKAA2*; (**d**–**f**) are *MSMB*. (**a**,**d**) Phylogenetic tree for the 10 species. (**b**,**e**) Motif structural analysis for the 10 species. (**c**,**f**) Significant motifs of *PRKAA2* and *MSMB* across the 10 species. The different colored letters signify the abbreviation of different amino acids.

**Figure 2 genes-14-00113-f002:**
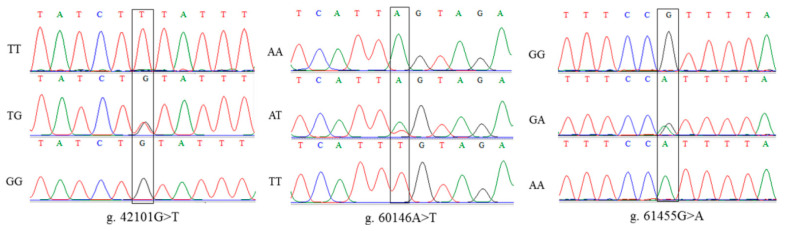
Sequencing results of three SNPs in *PRKAA2*.

**Figure 3 genes-14-00113-f003:**
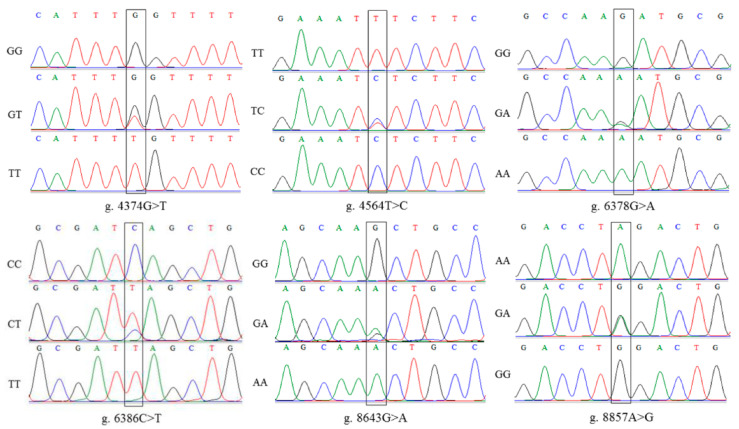
Sequencing results of six SNPs in *MSMB*.

**Figure 4 genes-14-00113-f004:**
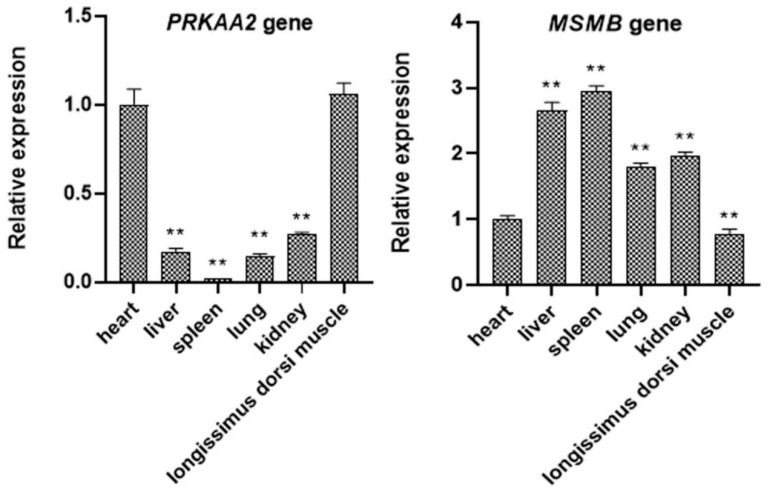
Expression of *PRKAA2* and *MSMB* in pig tissues. Note: The heart was used as a reference for all tissues. ** implies very significant difference (*p* < 0.01), and no * signifies no significant difference (*p* > 0.05).

**Table 1 genes-14-00113-t001:** Primers for porcine *PRKAA2*.

Primer Name	Primer Sequence (5′-3′)	Amplified Fragment Size (bp)	Annealing Temperature (°C)	Application
*PRKAA2*-1	F: TTAGACTGACCTTCGAGCAAGGTCG	973	61	Exon 1
	R: TCACAGAGCAGGGCACTGAAGTC			
*PRKAA2*-2	F: ATTTGTTCTTCAATAATGTATGACT	859	61	Exon 2
	R: AAAAATCCTGATATGCTAACTTGAA			
*PRKAA2*-3	F: GCTGGTTGTCTTCATCTTGGTATCA	581	55	Exon 3
	R: ATTACTGCAGAAGCAACCCCAACTT			
*PRKAA2*-4	F: CCAGGGTTTGAATTGGATCTATAGC	421	63	Exon 4
	R: GCAGCTAGTATCCTTCTAAACACCA			
*PRKAA2*-5	F: CAGCATATGGAAGTTCCCAGGCT	777	65	Exon 5
	R: TTTGATCACTGGCCTGAGGCAGT			
*PRKAA2*-6	F: TCAGGTATTGCCGTAGGGCTAGTTA	812	65	Exon 6
	R: GCCTAACATAGATCAGGCATTCAG			
*PRKAA2*-7	F: GCCACATACCCAAGTCTGAATAATC	823	59	Exon 7
	R: GCCAGAAGCATCTAGACCACTAAAT			
*PRKAA2*-8	F: ACAGTACCTGTTACTGTGCCAGGTT	533	63	Exon 8
	R: CTTCTCAGAGTTCATGCCTGCGT			
*PRKAA2*-9	F: TAGTGATGTCTGTTACGATTGAGGG	667	63	Exon 9
	R: CACCTAGTAAAGACACCGCCTATGT			
*PRKAA2*	F: GCCCAGTTACTTATTTCCT	189	60	Real-time PCR
	R: TTCATTATTCTCCGATTGTC			
*GAPDH*	F: TTGTGATGGGCGTGAACC	169	58	Reference gene
	R: GTCTTCTGGGTGGCAGTGAT			

Note: F stands for upstream primer and R stands for downstream primer.

**Table 2 genes-14-00113-t002:** Primers for porcine *MSMB*.

Primer Name	Primer Sequence (5′-3′)	Amplified Fragment Size (bp)	Annealing Temperature (°C)	Application
*MSMB*-1	F: GATGCACATGCCTGAAAGGACTC	668	55	Exon 1
	R: GAGTGATGGTTCCCAATTTGCTGA			
*MSMB*-2	F: GTCCAACAGATCCATATCAGCCTA	858	59	Exon 2
	R: CTGTCCCAACCCTTTCTCTCATATA			
*MSMB*-3	F: ATCACCCTAAATGCCCCTGACTCA	778	57	Exon 3
	R: TGAAAGAGGCATAGCTGTCCTTAG			
*MSMB*-4	F: CCTACCTGGAGTGACTGACACATA	676	61	Exon 4
	R: AGTTAGAGGCCTAGGGAATGAGG			
*MSMB*	F: AAAGAAGGACCCAGGAAAG	145	60	Real-time PCR
	R: CAATCATAGACAGTTAGAGGC			
*GAPDH*	F: TTGTGATGGGCGTGAACC	169	58	Reference gene
	R: GTCTTCTGGGTGGCAGTGAT			

Note: F stands for upstream primer and R stands for downstream primer.

**Table 3 genes-14-00113-t003:** Genetic diversity analysis of *PRKAA2* mutant groups.

SNPs	Genotypic Frequencies			Allelic Frequency		He	Ho	Ne	PIC	χ^2^
g.42101G>T	GG	GT	TT	G	T	0.498	0.502	1.991	0.374	1.833
	0.244 (40)	0.445 (73)	0.311 (51)	0.466	0.534					
g.60146A>T	AA	AT	TT	A	T	0.480	0.520	1.922	0.365	2.482
	0.390 (64)	0.421 (69)	0.189 (31)	0.601	0.399					
g.61455G>A	GG	GA	AA	G	A	0.499	0.501	1.996	0.375	0.649
	0.213 (35)	0.530 (87)	0.256 (42)	0.479	0.521					

Note: HE, heterogeneity; HO, pureness; NE, effective allele; PIC, polymorphic information content; PIC < 0.25 implies low polymorphism; 0.5 > PIC > 0.25 denotes moderate polymorphism; PIC > 0.5 suggests high polymorphism; χ^2^ < 5.99 denotes Hardy-Weinberg equilibrium; χ^2^ > 5.99 signifies Hardy-Weinberg disequilibrium.

**Table 4 genes-14-00113-t004:** Genetic diversity analysis of *MSMB* mutant groups.

SNPs	Genotypic Frequencies			Allelic Frequency		He	Ho	Ne	PIC	χ^2^
g.4374G>T	GG	GT	TT	G	T	0.485	0.515	1.943	0.368	3.488
	0.378 (62)	0.415 (68)	0.207 (34)	0.585	0.415					
g.4564T>C	TT	TC	CC	T	C	0.477	0.523	1.913	0.363	0.285
	0.378 (62)	0.457 (75)	0.165 (27)	0.607	0.393					
g.6378G>A	GG	GA	AA	G	A	0.499	0.502	1.994	0.374	0.014
	0.226 (37)	0.494 (81)	0.280 (46)	0.473	0.527					
g.6386C>T	CC	CT	TT	C	T	0.499	0.502	1.994	0.374	0.014
	0.226 (37)	0.494 (81)	0.280 (46)	0.473	0.527					
g.8643G>A	GG	GA	AA	G	A	0.491	0.509	1.965	0.371	0.518
	0.201 (33)	0.463 (76)	0.335 (55)	0.433	0.567					
g.8857A>G	AA	AG	GG	A	G	0.483	0.517	1.935	0.337	0.588
	0.152 (25)	0.512 (84)	0.335 (55)	0.409	0.591					

Note: HE, heterogeneity; HO, pureness; NE, effective allele; PIC, polymorphic information content; PIC < 0.25 implies low polymorphism; 0.5 > PIC > 0.25 denotes moderate polymorphism; PIC > 0.5 suggests high polymorphism; χ^2^ < 5.99 denotes Hardy-Weinberg equilibrium; χ^2^ > 5.99 signifies Hardy-Weinberg disequilibrium.

**Table 5 genes-14-00113-t005:** Association analysis of three SNPs in *PRKAA2* and growth trait.

SNPs	Genotype	W/kg	B S/cm	B H/cm	C C/cm	A C/cm	T C/cm	C D/cm	C W/cm	L H C/cm
g.42101G>T	GG	89.09 ± 3.99A	108.80 ± 4.91A	74.78 ± 4.16	109.83 ± 3.66A	119.20 ± 4.09A	19.78 ± 1.31A	39.38 ± 2.65	30.03 ± 2.13	71.40 ± 2.04
	GT	91.42 ± 4.35B	110.01 ± 4.59	75.21 ± 3.98	110.37 ± 3.31a	119.66 ± 3.75a	20.21 ± 1.18	40.14 ± 2.78	30.22 ± 1.95	71.51 ± 2.01
	TT	92.03 ± 4.81B	111.39 ± 4.42B	76.12 ± 3.81	111.96 ± 3.86bB	121.35 ± 3.92bB	20.59 ± 1.20B	40.51 ± 3.09	30.00 ± 1.84	71.69 ± 2.04
g.60146A>T	AA	91.67 ± 4.42	110.42 ± 4.38	75.88 ± 3.82	111.06 ± 3.46	119.86 ± 4.09	20.36 ± 1.17	40.41 ± 2.64a	29.94 ± 1.75	71.95 ± 1.68
	AT	90.37 ± 4.75	110.29 ± 4.83	75.33 ± 4.06	110.45 ± 4.00	120.16 ± 3.86	20.16 ± 1.39	40.20 ± 3.10	30.10 ± 2.16	71.30 ± 2.16
	TT	91.24 ± 4.19	109.26 ± 5.05	74.48 ± 4.10	110.68 ± 3.24	120.32 ± 4.04	20.06 ± 1.06	39.06 ± 2.62b	30.45 ± 1.89	71.19 ± 2.24
g.61455G>A	GG	90.86 ± 4.21	109.97 ± 4.79	76.03 ± 3.78	111.09 ± 3.25	120.71 ± 3.62	20.43 ± 1.04	41.00 ± 2.77a	29.91 ± 2.02	71.37 ± 1.94
	GA	90.40 ± 4.65a	109.84 ± 4.78	75.33 ± 4.03	110.69 ± 4.00	120.02 ± 4.09	20.05 ± 1.31	39.64 ± 3.01b	30.11 ± 1.92	71.59 ± 2.19
	AA	92.52 ± 4.32b	110.93 ± 4.44	74.95 ± 4.09	110.52 ± 3.24	119.64 ± 4.01	20.40 ± 1.25	40.17 ± 2.47	30.24 ± 2.01	71.57 ± 1.74

Note: W: Weight/kg, B S: Body straight/cm, B H: Body height/cm, C C: Chest circumference/cm, A C: Abdominal circumference/cm, T C: Tube circumference/cm, C D: Chest depth/cm, C W: Chest width/cm, L H C: Leg and hip circumference/cm. a, b indicate significant differences between different genotypes (*p* < 0.05); A, B indicate extremely significant differences between different genotypes (*p* < 0.01).

**Table 6 genes-14-00113-t006:** Association analysis of six SNPs in *MSMB* and growth traits.

SNPs	Genotype	W/kg	B S/cm	B H/cm	C C/cm	A C/cm	T C/cm	C D/cm	C W/cm	L H C/cm
g.4374G>T	GG	91.41 ± 4.76a	110.05 ± 5.01	74.02 ± 3.94aA	110.48 ± 3.61	120.60 ± 3.95	20.05 ± 1.26	39.19 ± 2.95aA	30.34 ± 2.13	71.21 ± 2.16
	GT	91.59 ± 4.21a	110.31 ± 4.31	76.29 ± 3.80B	110.71 ± 3.57	119.41 ± 4.15	20.35 ± 1.26	40.62 ± 2.65B	29.84 ± 1.84	71.72 ± 1.91
	TT	89.27 ± 4.43b	110.00 ± 4.96	76.06 ± 3.82b	111.24 ± 3.93	120.44 ± 3.49	20.26 ± 1.21	40.56 ± 2.81b	30.21 ± 1.84	71.76 ± 1.95
g.4564T>C	TT	91.41 ± 4.76	110.05 ± 5.01	74.02 ± 3.94aA	110.48 ± 3.61	120.60 ± 3.95	20.05 ± 1.26	39.19 ± 2.95aA	30.34 ± 2.13	71.21 ± 2.16
	TC	91.31 ± 4.35	110.15 ± 4.32	76.29 ± 3.80B	110.72 ± 3.45	119.47 ± 4.05	20.39 ± 1.24	40.57 ± 2.61B	29.93 ± 1.80	71.76 ± 1.90
	CC	89.44 ± 4.32	110.37 ± 5.12	76.00 ± 3.82b	111.33 ± 4.29	120.56 ± 3.62	20.15 ± 1.23	40.67 ± 2.95b	30.04 ± 1.97	71.67 ± 1.98
g.6378G>A	GG	90.48 ± 4.61	110.38 ± 4.95	75.11 ± 4.38	110.65 ± 3.80	120.11 ± 3.56	20.19 ± 1.27	39.19 ± 3.15a	30.16 ± 2.02	71.76 ± 2.50
	GA	91.62 ± 4.32	110.01 ± 4.58	75.48 ± 3.98	110.88 ± 3.62	120.28 ± 4.40	20.28 ± 1.25	40.40 ± 2.84b	30.04 ± 1.93	71.54 ± 1.91
	AA	90.48 ± 4.81	110.20 ± 4.78	75.43 ± 3.73	110.54 ± 3.65	119.67 ± 3.49	20.13 ± 1.26	40.20 ± 2.55	30.17 ± 1.99	71.35 ± 1.79
g.6386C>T	CC	90.48 ± 4.61	110.38 ± 4.95	75.11 ± 4.38	110.65 ± 3.80	120.11 ± 3.56	20.19 ± 1.27	39.19 ± 3.15a	30.16 ± 2.02	71.76 ± 2.50
	CT	91.62 ± 4.32	110.01 ± 4.58	75.48 ± 3.98	110.88 ± 3.62	120.28 ± 4.40	20.28 ± 1.25	40.40 ± 2.84b	30.04 ± 1.93	71.54 ± 1.91
	TT	90.48 ± 4.81	110.20 ± 4.78	75.43 ± 3.73	110.54 ± 3.65	119.67 ± 3.49	20.13 ± 1.26	40.20 ± 2.55	30.17 ± 1.99	71.35 ± 1.79
g.8643G>A	GG	90.22 ± 4.08	109.76 ± 4.66	74.97 ± 3.80	111.03 ± 4.07	121.18 ± 4.19a	20.00 ± 1.27	40.39 ± 2.89	30.12 ± 2.07	71.48 ± 2.35
	GA	90.75 ± 4.77	109.87 ± 4.32	74.84 ± 4.14a	110.20 ± 3.64	119.33 ± 4.22b	20.29 ± 1.27	39.89 ± 3.04	30.22 ± 2.00	71.63 ± 2.05
	AA	91.94 ± 4.40	110.76 ± 5.21	76.38 ± 3.73b	111.29 ± 3.34	120.44 ± 3.26	20.25 ± 1.21	40.11 ± 2.61	29.93 ± 1.84	71.44 ± 1.79
g.8857A>G	AA	90.58 ± 4.21	109.68 ± 4.63	75.28 ± 3.74	111.36 ± 4.20	121.52 ± 4.23a	20.08 ± 1.22	40.84 ± 2.88	30.16 ± 1.84	72.04 ± 1.90
	AG	90.59 ± 4.68	109.88 ± 4.36	74.76 ± 4.12a	110.18 ± 3.63	119.40 ± 4.19b	20.24 ± 1.30	39.81 ± 3.00	30.20 ± 2.10	71.45 ± 2.19
	GG	91.94 ± 4.40	110.76 ± 5.21	76.38 ± 3.73b	111.29 ± 3.34	120.44 ± 3.26	20.25 ± 1.21	40.11 ± 2.61	29.93 ± 1.84	71.44 ± 1.79

Note: W: Weight/kg, B S: Body straight/cm, B H: Body height/cm, C C: Chest circumference/cm, A C: Abdominal circumference/cm, T C: Tube circumference/cm, C D: Chest depth/cm, C W: Chest width/cm, L H C: Leg and hip circumference/cm. a, b indicate significant differences between different genotypes (*p* < 0.05); A, B indicate extremely significant differences between different genotypes (*p* < 0.01).

## Data Availability

Not applicable.

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
