# Peer review of "Association Analysis of PRKAA2 and MSMB Polymorphisms and Growth Traits of Xiangsu Hybrid Pigs"

_genes, 2022, doi:10.3390/genes14010113_

Round 1

Reviewer 1 Report

I reviewed the article titled: “Relationship between the tissue expression and polymorphism of PRKAA2 and MSMB and growth traits of Xiangsu hybrid pigs” and I found it very interesting, however not very well-prepared. The authors suggested in the title the relationship between some growth traits with tissue expression of two genes, yet there was no information regarding such relationship in the manuscript. Therefore, the aim of the study and the results did not correspond with the title. In my opinion the manuscript should be extensive reviewing the following points:

Abstract:

[14] Variants should be described according to the HGVS nomenclature (see website: https://varnomen.hgvs.org/recommendations/general/), therefore it should be i.e. c.42101G>T. Please correct it throughout the whole manuscript.

[18] Please indicate some of these growth traits described in the manuscript for the information for the other authors.

Introduction:

[61-62] If the MSMB gene was not linked with animal growth, why was it chosen in this study? Please explain in more detail.

[63-64] The aim of the study does not correspond with the title. The authors suggest in the title the analysis of the SNPs and expression profile in terms of growth traits, whereas the aim of study is linked solely to dominant genes and genotypes. Please change either the title or the aim of the study.

Materials and Methods:

[84] Please provide a table with an information about these pigs, like sex, mean values and standard deviation of these growth traits etc.

[88] Which three animals were sacrificed? They were chosen randomly or based on some features? Why only three?

[103] The authors used only one reference gene – GAPDH. Was it enough to determine relative mRNA abundance for the whole transcriptome? Did the authors verify other genes or select this gene based on the literature?

[104] Are these primers overlapping all the exon gene sequence or only parts of these exons?

Results:

[193] "The", typo

[199] In my opinion tables 5 and 6 are not very clear. I would suggest to present the results without approximation "±3.994". Moreover, I would change from "a,b" to "*", and from "A,B" to "**" or I would indicate the significant results with bold font. The differences between groups are visible based on the values, therefore in my opinion there's no need to separate it with a,b sybmols. In case of significant and extremely significant differences like: 121.353±3.918bB I would suggest to present only extremely significant results.

Discussion:

[238] It is supposed to be g.32832382 according to the HGVS nomenclature

[239] It is supposed to be g.32832382G>A

Conclusions:

[270-272] This sentence is a result of the research and is based solely on three samples. It is not strictly linked with the aim of the study.

Author Response

Point 1: The authors suggested in the title the relationship between some growth traits with tissue expression of two genes, yet there was no information regarding such relationship in the manuscript. Therefore, the aim of the study and the results did not correspond with the title.

Response 1: For your comments, we have rewritten title (Association analysis of PRKAA2 and MSMB polymorphisms and growth traits of Xiangsu hybrid pigs)

Abstract:

Point 2: [14] Variants should be described according to the HGVS nomenclature (see website: https://varnomen.hgvs.org/recommendations/general/), therefore it should be i.e. c.42101G>T. Please correct it throughout the whole manuscript.

Response 2: For your opinion, we have rewritten the name of the mutation site. We followed the genome published by the NCBI, so all the names are g.

Point 3: [18] Please indicate some of these growth traits described in the manuscript for the information for the other authors.

Response 3: For your comments, We have revised it in the original manuscript.

Introduction:

Point 4: [61-62] If the MSMB gene was not linked with animal growth, why was it chosen in this study? Please explain in more detail.

Response 4: For your comments, MSMB gene has not been found to be associated with growth traits. However, we only took it as a candidate gene at that time, and the final analysis result showed that MSMB gene was related to the growth traits of Xiangsu hybrid pigs.

Point 5: [63-64] The aim of the study does not correspond with the title. The authors suggest in the title the analysis of the SNPs and expression profile in terms of growth traits, whereas the aim of study is linked solely to dominant genes and genotypes. Please change either the title or the aim of the study.

Response 5: For your comments, we have rewritten title.

Materials and Methods:

Point 6: [84] Please provide a table with an information about these pigs, like sex, mean values and standard deviation of these growth traits etc.

Response 6: For your comments. The experimental pigs were castrated and depopulated separately after birth, therefore, this experiment did not distinguish between gilts and sows.

Point 7: [88] Which three animals were sacrificed? They were chosen randomly or based on some features? Why only three?

Response 7: Thank you for your comments. Three Xiangsu hybrid pigs were sacrificed, belonging to chosen randomly. Because one is about the requirement in ethics to minimize slaughter while satisfying the test conditions, and the second is that the sample size of 3 pigs per group can meet the statistical requirements according to the statistical requirements.

Point 8: [103] The authors used only one reference gene –GAPDH. Was it enough to determine relative mRNA abundance for the whole transcriptome? Did the authors verify other genes or select this gene based on the literature?

Response 8: For your comments. GAPDH has long been used as a reference gene in pigs and has become more representative, while other reference genes are not as widely used as GAPDH.

Point 9: [104] Are these primers overlapping all the exon gene sequence or only parts of these exons?

Response 9: For your comments. These primers overlapping only parts of these exons.

Results:

Point 10: [193] "The", typo

Response 10: Thank you for your advice. We have revised it in the original manuscript.

Point 11: [199] In my opinion tables 5 and 6 are not very clear. I would suggest to present the results without approximation "±3.994". Moreover, I would change from "a,b" to "*", and from "A,B" to "**" or I would indicate the significant results with bold font. The differences between groups are visible based on the values, therefore in my opinion there's no need to separate it with a,b sybmols. In case of significant and extremely significant differences like: 121.353±3.918bB I would suggest to present only extremely significant results.

Response 11: Thank you for your suggestion. Based on your comments and those of other reviewers, we have addressed Tables 5 and 6. ± Cannot be removed as this is the way to know the data discrepancies.

Discussion:

Point 12: [238] It is supposed to be g.32832382 according to the HGVS nomenclature.

Response 12: Thank you for your advice. We have revised it in the original manuscript.

Point 13: [239] It is supposed to be g.32832382G>A.

Response 13: Thank you for your advice. We have revised it in the original manuscript.

Conclusions:

Point 14: [270-272] This sentence is a result of the research and is based solely on three samples. It is not strictly linked with the aim of the study.

Response 14: Thank you for your advice. We have revised it in the original manuscript.

Reviewer 2 Report

This manuscript reports work performed to test the hypothesis that SNPs in the PRKAA2 and MSMB genes are potential selectable markers in the Xiangsu hybrid pig.  The authors tested this hypothesis by apparently using a direct PCR cloning approach to sequence exons for each gene and then determining if allelic frequencies of PRKAA2 and MSMB correlated with morphometric traits measured in this breed.  Generally the subject matter should be of interest to the general readership of Genes.  However, several issues exist with the current manuscript that lessen its clarity and impact. Please consider:

1) In general, there is not enough detail provided in the Materials and Methods section to allow proper evaluation of the approach and results. For instance, it appears that direct PCR cloning was utilized to obtain the reported sequence data pertaining to the exons however the majority of SNPs identified were intronic. The amplicon sizes appear to be problematic as many appear to contain more nucleotides than the typical error rate for Taq polymerases raising the issue that the amplication itself may have confounded the results by introducing copy errors. This is a concern regardless of amplicon length. The only controls described involved the use of gel electrophoresis to subjectively verify amplicon size.

2) A figure mapping the exon/introns for each gene and the placement of primers would be very helpful.

3) The value of figures 1 and 4 is unclear relative to the hypothesis/objectives of this work.

4) It is unclear why the current approach was adopted versus sequencing the entire genome which has become very cost efficient given the emergence of new sequencing technologies. 

5) No sequencing data set was provided either in the manuscript of supplemental materials.

6) The phenotypic traits appear morphometric rather than related to growth performance or physiology.  It is unclear how the reported associations would be used as selection criterion to improve the breed’s utility. It is unclear why important performance traits such as rate of gain, efficiency of gain, voluntary feed intake, body composition etc were not measured.

7) Primer efficiencies were not reported for the real-time PCR assay. It is unclear if error bars denote standard deviations or standard errors of the mean. Regardless, the reported error is exceptionally small for real-time PCR data especially given only 6 apparent replicates were utilized. The description in the methods is insufficient to allow understanding of the approach. Also, cycle numbers need to be reported so that readers can better understand level of expression. Were standard curves performed?  Were primers intron-spanning?  How was the assay validated? No controls were reported.

Author Response

Point 1: In general, there is not enough detail provided in the Materials and Methods section to allow proper evaluation of the approach and results. For instance, it appears that direct PCR cloning was utilized to obtain the reported sequence data pertaining to the exons however the majority of SNPs identified were intronic. The amplicon sizes appear to be problematic as many appear to contain more nucleotides than the typical error rate for Taq polymerases raising the issue that the amplication itself may have confounded the results by introducing copy errors. This is a concern regardless of amplicon length. The only controls described involved the use of gel electrophoresis to subjectively verify amplicon size.

Response 1: Thank you for your suggestion. Although the specificity of some primers in the picture is poor, it does not affect the Sanger sequencing results. Also, the second picture is 2 genes, the first one is MSMB gene and the later one is not.

Point 2: A figure mapping the exon/introns for each gene and the placement of primers would be very helpful.

Response 2: Thank you for your suggestion. We have provided.

Point 3: The value of figures 1 and 4 is unclear relative to the hypothesis/objectives of this work.

Response 3: Thank you for your suggestion. Figure 1 shows a partial bioinformatics analysis of these two genes. Figure 4 shows the detection of the relative expression of these two genes in several tissues.

Point 4: It is unclear why the current approach was adopted versus sequencing the entire genome which has become very cost efficient given the emergence of new sequencing technologies.

Response 4: Thank you for your suggestion. First, this paper mainly wanted to examine the effect of this candidate gene on growth, rather than an experiment designed from a holistic histological perspective. Secondly, due to financial constraints, the histology could not be done.

Point 5: No sequencing data set was provided either in the manuscript of supplemental materials.

Response 5: Thank you for your suggestion. The above document is currently a confidential document due to the data in the declared strains, but I have provided some of it in the supplemental materials.

Point 6: The phenotypic traits appear morphometric rather than related to growth performance or physiology. It is unclear how the reported associations would be used as selection criterion to improve the breed’s utility. It is unclear why important performance traits such as rate of gain, efficiency of gain, voluntary feed intake, body composition etc were not measured.

Response 6: Thank you for your suggestion. My main research is on the growth traits of Xiangsu hybrid pigs, and the indicators you mentioned are not my main research scope.

Point 7: Primer efficiencies were not reported for the real-time PCR assay. It is unclear if error bars denote standard deviations or standard errors of the mean. Regardless, the reported error is exceptionally small for real-time PCR data especially given only 6 apparent replicates were utilized. The description in the methods is insufficient to allow understanding of the approach. Also, cycle numbers need to be reported so that readers can better understand level of expression. Were standard curves performed? Were primers intron-spanning? How was the assay validated? No controls were reported.

Response 7: Thank you for your suggestion. Firstly, the primers were not designed in a way to find a suitable position across the intron region. Secondly, this experiment was only to detect the expression of mRNA of PRKAA2 gene and MSMB gene in several tissues of Xiangsu hybrid pigs, and no standard curve was made, again, I apologize.

Reviewer 3 Report

Lines 124-134 the software version and the reference of the software may be given

Line 138 "for the 10 mammalian or domestic animals included in the study: pig (S. scrofa), cattle " may be modified as " with ten mammalian or domestic animals viz.,  pig (S. scrofa), cattle.............."

Table 5 and 6 : Kindly indicate the values after mean is SE or Sd and also convert all the three decima to two decimal format.

Lines 252-263: need discussion in relation to the earlier published report

In general the article is well written and need more focused discussion for better conveying of the meaning

Author Response

Point 1: Lines 124-134 the software version and the reference of the software may be given

Response 1: Thank you for your suggestion, we have revised it in the manuscript.

Point 2: Line 138 "for the 10 mammalian or domestic animals included in the study: pig (S. scrofa), cattle " may be modified as " with ten mammalian or domestic animals viz., pig (S. scrofa), cattle.............."

Response 2: Thank you for your suggestion, we have revised it in the manuscript.

Point 3: Table 5 and 6 : Kindly indicate the values after mean is SE or Sd and also convert all the three decima to two decimal format.

Response 3: Thank you for your suggestion, we have revised it in the manuscript.

Point 4: Lines 252-263: need discussion in relation to the earlier published report.

Response 4: Thank you for your suggestion, we have revised it in the manuscript.

Point 5: In general the article is well written and need more focused discussion for better conveying of the meaning

Response 5: Thank you again for your advice and support.

Reviewer 4 Report

Xu et al., used in vitro and bioinformatics platforms to study the tissue expression of PRKAA2 and MSMB genes for growth traits in Xiangsu hybrid pigs. With the present form, the manuscript needs editing especially motivation for picking those two genes to study the growth traits. It is also not clear why authors had to use only 3 out of 164 to study animal weight and body size. Also, authors must remove all sort of grammatical errors.

Author Response

Point 1: Xu et al., used in vitro and bioinformatics platforms to study the tissue expression of PRKAA2 and MSMB genes for growth traits in Xiangsu hybrid pigs. With the present form, the manuscript needs editing especially motivation for picking those two genes to study the growth traits. It is also not clear why authors had to use only 3 out of 164 to study animal weight and body size. Also, authors must remove all sort of grammatical errors.

Response 1: Thank you for your suggestion. These results suggest that PRKAA2 and MSMB can be used in marker-assisted selection to improve growth related traits in Xiangsu hybrid pigs. Three pigs were selected for slaughter. Because one is about the requirement in ethics to minimize slaughter while satisfying the test conditions, and the second is that the sample size of 3 pigs per group can meet the statistical requirements according to the statistical requirements.

Round 2

Reviewer 1 Report

The article can now be published.

Author Response

第 1 点:文章现在可以发表了。

回复1:感谢您的支持和认可。

Reviewer 2 Report

Accept in present form

Author Response

Point 1: Accept in present form.

Response 1: Thank you for your support and recognition.

Reviewer 4 Report

I recommend the authors to set the Motivation  thoroughly in the manuscript. Later, it can be accepted

Author Response

Point 1: I recommend the authors to set the Motivation thoroughly in the manuscript. Later, it can be accepted.

Response 1: Thank you for your suggestion. For your comments, We have revised it in the original manuscript.
